# Psycho-Behavioural Segmentation in Food and Nutrition: A Systematic Scoping Review of the Literature

**DOI:** 10.3390/nu13061795

**Published:** 2021-05-25

**Authors:** Eva L. Jenkins, Samara Legrand, Linda Brennan, Annika Molenaar, Mike Reid, Tracy A. McCaffrey

**Affiliations:** 1Department of Nutrition, Dietetics and Food, Monash University, Notting Hill 3168, Australia; eva.jenkins@monash.edu (E.L.J.); samara.legrand@monash.edu (S.L.); annika.molenaar@monash.edu (A.M.); 2School of Media and Communication, RMIT University, Melbourne 3000, Australia; linda.brennan@rmit.edu.au; 3School of Economics, Finance and Marketing, RMIT University, Melbourne 3000, Australia; mike.reid@rmit.edu.au

**Keywords:** social marketing, segmentation, nutrition, food, psycho-behavioural variables

## Abstract

Inadequate dietary intakes are a key modifiable risk factor to reduce the risk of developing non-communicable diseases. To encourage healthy eating and behaviour change, innovative public health interventions are required. Social marketing, in particular segmentation, can be used to understand and target specific population groups. However, segmentation often uses demographic factors, ignoring the reasons behind why people behave the way they do. This review aims to explore the food and nutrition related research that has utilised psycho-behavioural segmentation. Six databases from were searched in June 2020. Inclusion criteria were: published 2010 onwards, segmentation by psycho-behavioural variables, outcome related to food or nutrition, and healthy adult population over 18 years. 30 studies were included; most were quantitative (*n* = 28) and all studies used post-hoc segmentation methods, with the tools used to segment the population varying. None of the segments generated were targeted in future research. Psycho-behavioural factors are key in understanding people’s behaviour. However, when used in post-hoc segmentation, do not allow for effective targeting as there is no prior understanding of behaviours that need to change within each segment. In future, we should move towards hybrid segmentation to assist with the design of interventions that target behaviours such as healthy eating.

## 1. Introduction

Suboptimal diets are a significant risk factor for non-communicable diseases, with The Global Burden of Disease study finding that mortality rates from suboptimal diets led to an estimated 11 million deaths and 255 million disability-adjusted life-years in 2017 alone [1]. Contemporary living environments contribute to poor dietary choices by increasing the accessibility and exposure to inexpensive, energy dense, nutrient poor but highly palatable food and beverages [2,3]. These environments, paired with technology advances, transport, and community structures that decrease physical activity have further contributed to poor health outcomes [3,4]. In addition to the physical environment, individuals also face numerous self-perceived barriers to healthy eating including; price (i.e., the perception that healthy foods are too expensive), taste, time constraints (e.g., lack of time to plan, shop, prepare and cook healthy foods), lack of motivation, as well as family and peer influences [5,6]. The implementation of interventions that promote healthy eating behaviours such as high intakes of fruits, vegetables, and wholegrains are complex, generating an increasing demand for health professionals to find new and innovative ways to encourage behavioural change [6,7].

To reduce non-communicable diseases and their associated modifiable risk factors, strategies that support healthy eating and encourage sustained behaviour change are required. Social marketing is often used to create behavioural change [8]. Social marketing is an ethically-guided practice that seeks to develop and integrate traditional marketing concepts with other approaches to influence behaviours that benefit individuals and communities for the greater social good [8]. In terms of social marketing interventions designed to change behaviours, segmentation has been shown to be valuable in understanding how to influence healthy eating behaviours [9,10].

Segmentation is based on the notion that individuals have different needs, preferences, and motivations to behave, and therefore different segments require different approaches to communications and interventions in order to persuade them to change behaviours [9,10,11,12]. The historical use of one-size-fits-all approach to campaigns and interventions, where the needs of different segments have not been accounted for in behaviour change strategies, messages and media choices, has often served to undermine the effectiveness of intervention efforts, and has left large numbers of individuals in different segments, unreached, uninterested, or unchallenged [13,14]. By dividing populations into smaller subgroups based on similarities, each group or segment can be targeted with a more customised message and media strategy designed to increase the effectiveness of interventions [9]. Broadly, there are a number of bases for segmentation including the use of psychographic, demographic, geographic, and behavioural factors (see Appendix A) [11,15].

Demographic and geographic variables have previously been used by researchers to segment target populations; however, research suggests that often these segments fail to effectively understand and predict behaviours [16]. Whilst demography and geography are important in the planning and design of interventions (e.g., deciding where to allocate funds or which communication channel to use), focusing on these variables alone means that information about how and why the target audience behaves as it does is often overlooked [16]. Predicting and supporting behaviour change requires an understanding of what motivates people and the social and economic influences on their behaviour [17,18]. Segmentation of a population by demographics alone is therefore less effective than the use of psycho-behavioural variables, which combine both behavioural and psychographic factors [19].

Psychographic segmentation explores the personality traits, beliefs, values, interests and lifestyles to understand motives of actions and behaviours—i.e., the ‘why’ and ‘what’ as opposed to the ‘who’ and ‘where’ provided by demographic and geographic segmentation [15,20]. Behavioural segmentation includes the frequency and quantity of behaviours such as purchasing, usage and consumption of certain foods [11,15]. Psychographic and behavioural variables can be employed as a primary basis for segmentation analysis to provide a more explanatory and predictive approach [18]. These variables can help to understand the underlying social and psychological factors that lead to certain behaviours in addition to observable demographic and geographic variables [18]. Delivering insights into what people think and understand and what motivates behaviours offers the potential to improve prediction of behaviours in addition to extending insights that can be used during program planning and design. Previous studies suggest combining segmentation variables (demographics, psychographics, geographic and behavioural variables) to provide a meaningful profile of the population [15,18]. 

In health promotion, segmentation is important because while the audience may be patients or clients, the principles of reaching and engaging them remain the same, it is the context of use that is different [21]. Some foundations of segmentation can be ascertained prior to planning a campaign, or a-priori, whereas others cannot. Understanding demographics will permit insight into factors such as age or income that influence a person’s health status [22] and are usually ascertained before planning a campaign (a-priori; See Appendix A for examples). Similarly, geographic segmentation allows health promoters to understand where people are located and to apply efforts to the right locations [18] and can also be done in advance (a-priori). On the other hand, psychographic segmentation enables the health promoter to connect more closely with the audience by creating communications that are specific to their belief structures [15]. However, these variables are not easily ascertained in advance of campaign design and therefore are often evaluated in a two-step or hybrid system. Firstly, deciding a qualitative description of segments, and secondly after data collection, using post-hoc analysis to evaluate and/or quantify the segments [15]. Similarly, segmenting using behavioural variables requires prior understanding of behaviours, hence is either done as a single qualitative stage or a two-stage hybrid process of segmentation. 

To our knowledge, there are a limited number of studies applying psycho-behavioural segmentation in the context of food and nutrition. Therefore, the objective of this scoping review is to explore the food and nutrition related research that has used psycho-behavioural segmentation in adults, by asking the research question “How is psycho-behavioural segmentation used in food and nutrition-related research?” The overarching aim of this paper is to provide formative research to determine the breadth and scope of segmentation within the contemporary food and nutrition environment. A systematic scoping review was deemed most appropriate for this purpose [23].

## 2. Materials and Methods

### 2.1. Search Strategy & Databases

The Preferred Reporting Items for Systematic Review (PRISMA) 2020 statement [24] and Scoping Review checklist [25] were used throughout the review process. Six key databases from science, marketing, and psychology were used to conduct the final search in conjunction with a University librarian on the 30 June 2020. Scopus (2770 results), OVID Medline (2696 results), CINAHL plus (1495 results), Emerald Insight (1556 results), PsycINFO (835 results), and Business Source Complete (124 results) were searched for title, abstract, and keywords. Search terms were related to segmentation and food and nutrition (see Appendix A). The final search yielded 6228 results once duplicates were removed (Figure 1). All articles were imported into Covidence Online Software (Veritas Health Innovation, Melbourne, Australia) to manage the reviewing process. 

### 2.2. Eligibility Criteria 

Inclusion criteria were: segmentation by psycho-behavioural variables, outcome related to food or nutrition, and healthy adult populations over 18 years old. As technology is a fast-changing field, a temporal limitation was put in place to include only research studies published between 2010 and 30 June 2020. Exclusion criteria were: segmentation by any of the following in isolation—dietary intake, or behavioural, demographic, anthropometric, or biochemistry variables, segmentation by small area analysis, populations under 18 years old, papers about alcohol consumption only, papers about purchase intention or purchasing specific foods only (e.g., functional foods), papers about purchasing experience (e.g., food labelling), papers focussing on eating disorders, experimental studies, and grey literature. 

### 2.3. Screening

Three investigators (A.M., E.L.J., S.L.) independently screened the title and abstracts of included papers against the inclusion and exclusion criteria and repeated the process for full-text screening, with each paper requiring two investigators to screen it. All conflicts were discussed until a joint consensus was reached (inter-rater reliability 0.41 for abstract screening; 0.76 for full-text screening). There were 27 final papers (Figure 1).

### 2.4. Data Extraction

Data extraction was conducted by two researchers (E.L.J., S.L.) using Microsoft Excel (2019, Microsoft Corporation, Redmond, WA, USA) with all extracted data being cross-checked before collating (see Appendix A). The main focuses of data extraction were demographic data of the study population, basis for segmentation, segmentation method and whether it was a-priori (pre-defined) or post-hoc (conducted after data collection), the number of segments generated and their names, key findings, and whether future research was conducted by the authors to target these segments. 

## 3. Results

### 3.1. Results

The final searches retrieved 9476 articles, with 79 papers remaining after duplicate removal and abstract screening (Figure 1). Following full-text screening, 27 final papers published between 2011 and 2020 were included in the review. Three papers included two studies each, therefore 30 studies were included in the review [26,27,28]. 

Many studies that met most of the exclusion criteria were excluded in full-text screening for segmenting consumers based on psychographic, demographic, or behavioural variables rather than psycho-behavioural [29,30,31]. For example, Schnettler et al. segmented their study population based on the Satisfaction with Food Related Life scale, which included statements such as ‘I am pleased with my food’ and ‘food and meals give me satisfaction in daily life’ which do not relate to the actual behaviour of eating, and therefore were not classified as psycho-behavioural factors [29]. 

The majority of included studies were quantitative (*n* = 28) [13,26,27,32,33,34,35,36,37,38,39,40,41,42,43,44,45,46,47,48,49,50,51,52,53], with only two being qualitative [28,54]. Study populations ranged from *n* = 73 to *n* = 3085 participants aged between 18 years and 85+ years. The majority of studies included both male and female participants (*n* = 28) [13,26,27,28,32,33,34,35,36,37,39,40,41,42,44,45,46,47,48,49,50,51,52,53,54], however one had females only [38], and one did not specify sex or gender [43]. The studies included were from a range of countries such as Australia (*n* = 4) [13,33,47,54], Chile (*n* = 3) [48,49,50], and Croatia (*n* = 2) [32,42], and were published in a range of journals from public health, nutrition, and social marketing disciplines.

The majority of studies (*n* = 22) recruited adult volunteers or did not specify whether participants were incentivised to complete the study [13,26,28,32,33,34,35,36,37,38,39,40,41,42,43,44,45,46,51,52], five studies paid or incentivised participants on completion of the study [27,47,53,54], and three studies recruited student samples from universities without stating whether they were incentivised or volunteers [48,49,50]. Inclusion criteria for the studies were variable, the most common criteria requiring participants to be at least 18 years old [26,34,35,37,42,44,46,47,51,52,53,54].

### 3.2. Theory 

Of the 30 included studies, nine referenced a theory or model that underpinned the rationale of their research [13,27,33,37,41,44,53,54]. One paper included two studies in separate populations with the same theory used across both studies [27]. The majority used a single theory, including Self-Determination Theory [27], Means-End Chain Theory [41], The Motivation, Opportunity, and Ability theoretical framework, the Theory of Planned Behaviour [33] and Hierarchy of Food Needs Model which applies the principles of Maslow’s Hierarchy of Needs [53,55]. Two studies referenced multiple theories [37,44]. One incorporated the related theories of the Theory of Reasoned Action and the Theory of Planned Behaviour from the rational economic model/cognitive model [37] and the other used both the Theory of Planned Behaviour and the Transtheoretical Model of Behaviour Change [44]. The most common theory and model referenced included the Theory of Planned Behaviour [33,37,44], and the Transtheoretical Model of Behaviour Change [44,54], both models of individual behaviour change [8]. Five studies used theory to understand attitudes and beliefs and how they influence eating styles, behaviours and food preferences [27,37,41,53], three studies used theory to determine the segmentation variables [13,33,44] and one used theory to inform the analysis [54].

### 3.3. Method of Segmentation 

All included studies used post-hoc segmentation, meaning segments were generated after data collection rather than pre-defined before the research study began. The number of segments varied between studies; most found three segments (*n* = 14) [26,28,33,36,38,40,41,45,46,47,49,50,54], followed by four (*n* = 10) [27,32,34,35,39,43,44,48,53], two (*n* = 5) [13,28,37,51,52], and five (*n* = 1; Table 1) [42]. A majority of the studies (*n* = 24) used cluster analysis as their method of segmentation [13,26,27,28,32,33,34,35,38,39,40,41,42,43,45,47,48,49,50,51,52,53] with the most common being two-step (*n* = 17) [13,26,27,33,39,40,41,42,47,48,49,50,51,52,53], followed by Ward’s method (*n* = 5) [32,34,35,38,43], k-means (*n* = 1) [45], and hierarchical complete linkage clustering (*n* = 1) [28]. Two studies each used qualitative methods [15,28] and latent class cluster analysis for segmentation [36,46], and one study each used factor analysis [37] and latent class analysis [44].

### 3.4. Segmentation Tools 

The variables used to conduct segmentation analysis varied between studies and depended on the instrument used to collect the data (Table 1). Most commonly, studies created their own questionnaire based on scales developed in the literature and then used a combination of variables from the scales to conduct segmentation analysis (*n* = 10) [13,33,37,39,44,47,49,50,51]. Other studies created and imputed their own items for segmentation without using an existing scale (*n* = 3; Table 1) [41,45,52]. The most commonly used existing tool for segmentation was the Food Choice Questionnaire (FCQ; *n* = 6) [56], which recognises the influence of many factors such as sensory appeal, health, convenience, and familiarity on an individual’s food choice [26,32,34,35,42]. The Three Factor Eating Questionnaire (TFEQ) [27,38] was used in three studies, and both the modified version of the Food-Related Lifestyle Instrument [57] and the Mealtime Functionality Questionnaire [32] were used in two studies [36,43]. Other tools used for segmentation were the Family Eating Habits Questionnaire [48], Food Choice Values Questionnaire [53], and the Health and Taste Attitude Scale [46] (Table 1).

### 3.5. Outcomes of Segmentation 

The reporting of segments differed between studies, with some creatively naming segments and giving them personas, for example ‘Rational, health conscious consumers’ [47] which provides a succinct but clear summary from the segment name alone. One study used ‘Saints’ and ‘Sinners’ as the segment names to correspond to people’s health beliefs [54]. Other studies reported segment names as ‘Segment 1’, ‘Segment 2’ and so on [35,40], which allowed for less nuanced understanding of the segments prior to thoroughly reading results (Table 1). 

Despite using the same tools for segmentation (e.g., FCQ), the names and descriptions of segments differed between studies. Of studies using the FCQ to segment the population (*n* = 6), two of six [32,35] reported a segment that was ‘indifferent’ to the food choice factors, and three studies [26,34] reported a segment that was concerned with each of the food choice factors. However, all other segments had limited similarities to each other and thus were not comparable. The Food-Related Lifestyle instrument used in two studies [36,43] did not produce similar findings, except one segment from each that were indifferent to food-related lifestyle factors. The most similar outcomes between studies were those using the TFEQ for segmentation (*n* = 3) [27,38]. Despite having a different number of segments generated from cluster analysis, the three studies using the TFEQ all found a segment with a low tendency to emotionally eat—the ‘rational’ segment (in two studies) [27] and the ‘conscious’ segment [38]. Similarly, all three studies found segments that had a high likelihood of emotional and uncontrolled eating and low cognitive restraint—the ‘susceptible’ and ‘struggling’ segments [27] and the ‘emotional and hedonic eaters’ [38]. These results emphasise that each study was individual and produced unique results with limited similarities, with the most common similarity between all studies being a segment that is ‘indifferent’—i.e., shows limited interest in food and nutrition (Table 1). A full description of the segments generated in each included paper can be found in Appendix A.

## 4. Discussion

### 4.1. Summary of Evidence 

The use of psycho-behavioural segmentation in food and nutrition-related research is key to both identifying the nuances between different groups in the population, and also understanding and targeting a desired audience for the purposes of behaviour change. The aim of this scoping review was to explore and provide a summary of the research that incorporates such segmentation in adult populations, to reflect segmentation within the contemporary food and nutrition environment. We found that all of the studies used post-hoc segmentation methods and the majority were quantitative. The variables and instruments used for segmentation were diverse depending on the aims of the study leading to an array of descriptions and names of segments. Few studies incorporated segmentation theory into their research and only one study to our knowledge [28] used the segments for further research. 

In commercial marketing settings, segmentation provides a framework for closely examining and analysing the unique ‘consumer’ behaviours that occur within each segment [58]. Segmentation is essential where people make active choices about whether to engage in specific behaviours in order to ensure that marketing or communication resources are spent to best effect [59]. There is very little point in spending resources communicating with an entire population if they are not the people whose behaviours need to change. Segmentation is premised on the notion of target marketing [60]. The metaphor of ‘target market’ is one often used in the commercial sector whereas in communications, the term used is target audience. Notwithstanding the use of the term market or audience, the metaphor goes something like this: the organization shoots an arrow (marketing mix or campaign) at the target (market) and is considered successful if the market responds as expected (according to agreed impartial measurable outcomes). A core element to this metaphor is the idea that if you know who your target is, you can aim appropriately (a-priori segmentation). Another element in this metaphor is the concept of the ‘connection’ that occurs when the arrow meets the target. Successful targeting is when the arrow embeds itself in the small circle in the middle of the target (the bullseye). Thus, the arrow is firmly attached within a relatively small pre-defined space. The act of segmentation is determining that pre-defined space. 

In health promotion settings, where campaign designers are often working with small budgets and wide-ranging public health issues, spending funds astutely is important. Consequently, the more that can be known about the target audience before designing a campaign, the more likely targeting will be successful and campaign objectives will be met. If behaviour change is the goal—as it often is—then we need insight into the behaviours we want to change. We need to deeply understand why people behave the way they do, more than we need to understand how old they are or whether they finished high school. Demographics are not variable at the campaign level, in that a campaign cannot change someone’s age or educational background, although a campaign might aspire to change their health status. While demographic elements like age do not allow an insight into behaviours, they do permit an understanding of where people are and how they can be contacted, and so facilitate the connection between the health promoter and the audience. 

All studies included in this review utilised post-hoc segmentation which determines the number and characteristics of the segments from data following the implementation of the methods [7]. This method can be helpful in the design of healthy eating campaigns to utilise social cognitive variables to obtain information on target populations which can be used in future health promotion efforts. As a majority of the studies in this review were quantitative, post-hoc segmentation allowed the segments to be created based on the psycho-behavioural variables determined from their questionnaires. One commonly utilised tool was the FCQ, used in six of the included papers. Using a pre-existing tool with post-hoc segmentation does not allow for effective targeting of the population and hence is not effective in behaviour change. There are three major reasons for this, explained in terms of our metaphor of shooting an arrow at the target population: (1) you do not know who will be in each segment prior to using the tool; (2) you are unaware of whether you have hit the arrow onto the target; and (3) you cannot change the trajectory of the arrow because you have no prior understanding of the populations’ behaviours in advance. Therefore, the campaign or intervention effectiveness cannot be ascertained because the objectives for each segment are not set in advance of intervention design, even if they are evaluated afterwards.

There are two other classifications of segmentation methods; a-priori and hybrid [7]. In a-priori segmentation, the number and type of segments are determined before data collection and often used in health-behaviour research to observe demographic variables such as age, gender, etc. [7]. A-priori segmentation is where the target is known in advance of the intervention or campaign design. Expectations of outcome behaviours are set in advance of campaign implementation, most often by way of setting specific and measurable objectives. A-priori segmentation can be qualitative or quantitative in nature and in commercial settings often includes psycho-behavioural as well as demographic and geographic variables. Hybrid segmentation employs a two-step approach where a-priori segmentation is first used, followed by post-hoc segmentation to further identify clusters [7]. By using a hybrid approach, researchers can use qualitative methods to first identify segments based on psycho-behavioural factors and then further compare the demographics to provide a more nuanced approach and increase understanding of the target audience, especially in relation to creating opportunities for connection. For example, knowing what people eat and how they engage with food enables insight into behaviours, but knowing how old they are and where they live provides insight into contact points that can be used to initiate behavioural change. 

Qualitative methods are inductive, allowing researchers to understand participant values, motives, and behaviours, generating original concepts and exploring themes rather than using a deductive approach with predetermined variables to investigate a population of interest [61,62]. A 2020 study by Terp et al. examined older patient’s knowledge, skills and behaviour in order to understand their attitudes towards managing their nutritional needs [63]. The study consisted of both qualitative and quantitative sections [63]. The qualitative analysis indicated a limited level of self-management to meet their nutritional needs, whereas the quantitative results did not indicate a limited level of self-management [63]. Whilst this study did not utilise segmentation, it provides an example of how qualitative methods can contribute different knowledge and provide valuable data that is often overlooked or underexplored in quantitative research. In regard to segmentation, a combination of qualitative and quantitative methods has the potential to yield deeper insights than a single variable segmentation method. However, it may also result in less effective targeting as each dimension or variable added to the approach narrows the required size of the bullseye accordingly. For example, if the population is *n* = 10,000, women over 50 within that population (*n* = 2000), women over 50 who exercise regularly (*n* = 250), women over 50 who exercise regularly, eat breakfast and love life (*n* = 25), and so on, with each variable added reducing the size of the target, sometimes to the point of meaninglessness. Therefore, the variables used for segmentation must be carefully considered prior to the research taking place. 

There are numerous statistical techniques to conduct segmentation analysis, with each technique having its own way of interpreting the data [64]. To determine the segments, the majority of studies utilised quantitative methods and applied cluster analysis techniques to create homogenous groups based on the participants’ answers to scaled and closed ended questions. Unlike latent class analysis which can only use categorical data [64], cluster analysis can be applied to coded qualitative data within studies to determine the motives behind participants’ behaviours [65]. The majority of these statistical techniques are applied post-hoc, and consequently they cannot be used to assess campaign or intervention effectiveness. They can, at best, be used to describe audience characteristics after the event. 

The most common similarity between all included studies was an ‘indifferent’ segment, which expressed no interest in food and nutrition [32,35,43,51]. This finding is mirrored in research by Brennan et al., where six segments were generated based on their interest in living a healthy lifestyle and their food choices, with one of the six segments being ‘blissfully unconcerned’ and not caring about the food that they consume [15]. Subgroups of the population that do not care about their health or diet represent a challenge for public health practitioners, as typically these groups are not motivated by the same messages that inspire others to change, due to a lack of motivation and self-efficacy [66]. Unmotivated groups are unlikely to change their behaviour even with access to more information, and typically require larger systemic changes to make the unhealthy behaviour more difficult than the healthy behaviour [66]. For example, increasing the price of cigarettes and banning smoking in public places throughout Australia has made it highly inconvenient for smokers to continue their behaviour. In terms of healthy eating, having confectionery-free checkouts at the supermarket and healthy vending machines surrounding workplaces make purchasing energy-dense and highly palatable food and beverages less mindless and less convenient for consumers. These environmental changes could be effectively coupled with personalised health interventions arising from segmentation analysis to begin to motivate the indifferent consumers and generate long-term behaviour change. 

### 4.2. Gaps in Research 

Many of the studies included segmentation bases that are not effectively covered by the standard bases for segmentation such as demographic, geographic, or behavioural. For example, sensory appeals [26,32,35,39,42,52,53], physical [28] and hedonic [28,34,44] sensations do not fit into any of the existing categories, neglecting the important role that food plays in physiological terms. As such, the current bases for segmentation do not account for illnesses such as diabetes or autoimmune conditions such as coeliac disease. Understanding the physical and physiological dimensions of market segments are essential in developing meaningful and successful health promotion campaigns. Hence, the existing bases are applicable to commercial models of segmentation but more is needed for effective targeting in health settings. 

Most of the studies included in the review recognised that the segments generated in analysis would be beneficial to use in future research as they provided greater insight into the attitudes, motivations, and behaviours affecting different groups of the population. However, of the 30 included studies, only one did follow up research using the segments generated (to the authors’ knowledge). This represents a missed opportunity to use the segments in other population groups for more nuanced and targeted health promotion efforts. 

### 4.3. Limitations 

This review consisted of studies using participants’ self-reported data about their beliefs and behaviours, which introduces bias and the potential for manipulation of data, as participants may report what they believe to be the socially desirable answer [67]. The studies included all had different levels of detail in their reporting, with many missing demographic details of the segments or reporting incorrect percentages when describing their results. This made it difficult to collate, summarise, and compare the data. A methodological limitation of undertaking a scoping review is the omission of quality appraisal and risk of bias within the included studies, usually conducted during data extraction in a systematic literature review.

### 4.4. Implications and Future Directions

In future, it would be beneficial to use psycho-behavioural variables for segmentation rather than psychographic or behavioural alone. Psycho-behavioural data can lead to a more effective understanding of the population you are working with, meaning you can target their behaviours more effectively. Furthermore, moving from primarily post-hoc segmentation methods to hybrid methods, which combine both a-priori and post-hoc stages and typically involve a mix of qualitative and quantitative methods would allow for targeted and more effective health promotion. 

## 5. Conclusions

Targeted health promotion efforts are required to reduce the rates of non-communicable diseases including obesity and improve population health across the globe. Too often populations are segmented based on demographics e.g., age or income, rather than considerations of their thoughts, feelings, and behaviours, meaning the nuances between different population groups are overlooked. This review explored the use of psycho-behavioural segmentation in the context of food and nutrition, finding that the majority of research studies included used quantitative methods and post-hoc segmentation. Post-hoc segmentation is not effective in behaviour change efforts, as there is no prior knowledge of who will be in each segment or the behaviours that need to be addressed, and therefore no way of knowing whether you have effectively targeted the ‘right’ behaviours. In future, those using segmentation for health promotion should consider using hybrid methods if resources allow, whereby there is a two-step approach combining both a-priori and post-hoc methods. Hybrid approaches make space for both qualitative and quantitative stages and allow the consideration of both psycho-behavioural factors and demographic factors, leading to effective targeting of the desired population group and therefore a greater chance of behaviour change. 

## Figures and Tables

**Figure 1 nutrients-13-01795-f001:**
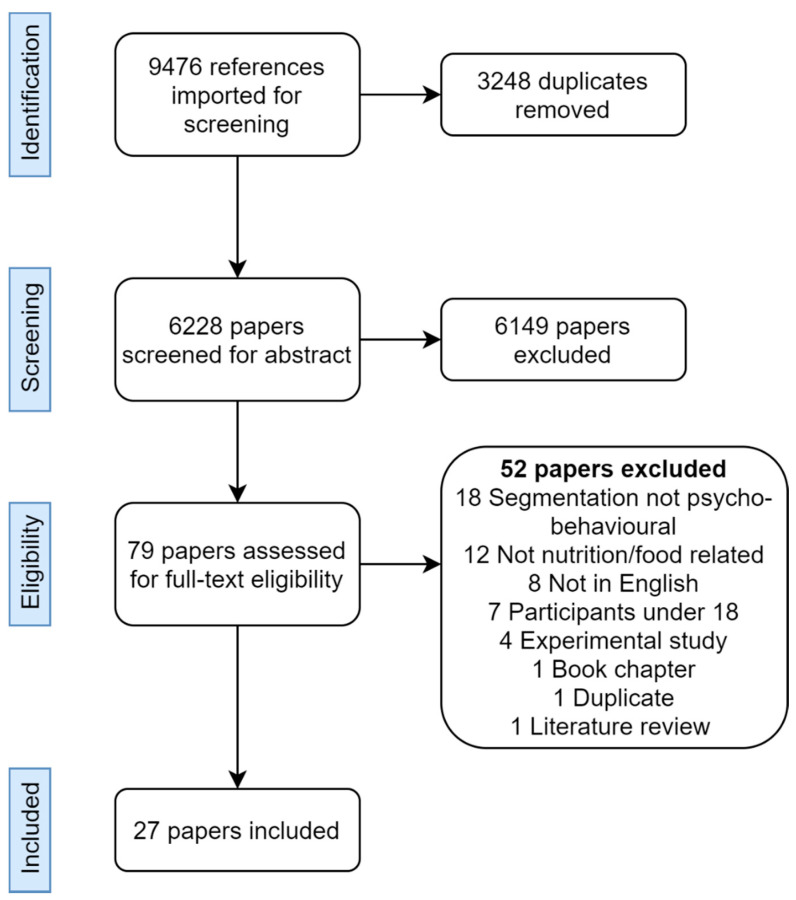
Preferred Reporting Items for Systematic Review (PRISMA) Scoping Review flow diagram, adapted from the 2020 PRISMA statement [24].

**Table 1 nutrients-13-01795-t001:** Included studies and the tools used for segmentation, number of segments generated and their names.

Author; Year; Location	Underlying Theory or Models	*n* Segmented; Age (Mean (SD) or Otherwise Specify); Gender/Sex ^a^	Tool Used for Segmentation *Variables/Question/Items Used*	Segmentation Method	No. Segments	Segment Number and Name (% of Sample)
Brečić et al.; 2017; Croatia	N/R	500; 47.6 (SD N/R); 46.8% male, 53.2% female	Food Choice Questionnaire (modified). *Questions under 4 factors (health and sensory, price, digestion, and convenience).*	Cluster analysis; Ward’s method	4	S1: Healthy and tasty food—23.6% S2: Convenient consumers—26.9% S3: Concerned consumers—27.2% S4: Indifferent consumers—19.9%
Brennan et al.; 2020; Australia	Transtheoretical model of behaviour change	195; 21.0 (2); 39.0% male, 61.0% female	Themes associated with dietary behaviours and attitudes towards eating.	Qualitative thematic analysis	3	S1: Saints ^f^ S2: Sinners ^f^ S3: Person in the pew ^f^
Burton et al.; 2017; Australia	Theory of planned behaviour	1059; >18–61+ (range); 35.3% male, 64.7% female	Developed their own questionnaire based on scales previously used in the literature. *Questions from 2 scales: perceived cooking capability and perceived nutrition knowledge*.	Cluster analysis; 2 step	3	S1: Low confidence for nutrition knowledge and cooking capability—22.9% S2: Moderate confidence for nutrition knowledge and cooking capability—48.3% S3: High confidence for nutrition knowledge and cooking capability—28.8%
Cabral et al.; 2017; Cape Verde	N/R	Study 1: 433; 35.9 (6.4); 35.6% male ^a^, 64.4% female ^a^ Study 2: 119; 35.2 (17.7); 34.6% male ^a^, 65.4% female ^a^	Food Choice Questionnaire (Portuguese version). *Questions under 9 factors (nutrition and diet, sensory appeal, mood, wellbeing, convenience, price, familiarity, ethical concern, natural content).*	Cluster analysis; 2 step	3	S1: Healthy—Study 1 12%, Study 2 25% S2: Hedonists—Study 1 35.7%, Study 2 51% S3: Engaged—Study 1 52.3%, Study 2 24%
den Uijl et al. 2016; The Netherlands	N/R	Study 1: 392; 65.8 (5.9); 40.3% male ^a^, 59.7% female ^a^ Study 2: 40; 66.9 (4.8); 50.0% male ^a^, 50.0% female ^a^	Mealtime functionality questionnaire. *Questions under 13 constructs (hunger, habit, liking, cosiness, pleasure, energising, rewarding, healthiness, pleasing, calming, physical needs, thoughtless eating, and environmental awareness).*	Cluster analysis; hierarchical complete linkage	Study 1: 3; Study 2: 2	Study 1: S1: Cosy socialisers—28% S2: Physical nutritioners—39% S3: Thoughtless rewarders—33% Study 2: S1: Cosy socialisers—50% S2: Physical nutritioners—50%
Espinoza-Ortega et al.; 2016; Mexico	N/R	202; 18–60 (range); 42.5% male ^a^, 57.2% female ^a^	Food Choice Questionnaire (modified). *Questions under 10 factors (care for weight and health, social sensitivity, practicality, economic aspects, not industrialised, hedonism, traditionality, familiarity, no sugar).*	Cluster analysis; Ward’s method	4	S1: Traditional—20.1% S2: Healthy not committed—41.6% S3: Conscious—27.0% S4: Careless—11.3%
Gama et al.; 2018; Malawi	N/R	489; 18–50+ (range); 68% male, 32% female	Food Choice Questionnaire (modified). *Questions under 5 factors (mood, health, price and preparation convenience, sensory appeal, familiarity).*	Cluster analysis; Ward’s method	4	S1 ^d^—30.0% S2 ^d^—13.0% S3 ^d^—33.0% S4 ^d^—24.0%
Grunert et al.; 2011; China	N/R	479; 39 (1.8); 50.1% male, 49.9% female	Food-Related Lifestyle Instrument (modified). *Questions from 5 dimensions (ways of shopping, quality aspects, cooking methods, consumption situation, purchase motives).*	Latent class cluster analysis	3	S1: Concerned—45.0% ^e^ S2: Uninvolved—33.0% ^e^ S3: Traditional—21.0% ^e^
Gunden et al.; 2020; Turkey	Theory of Reasoned Action; Theory of Planned Behaviour	371; N/R for total sample; 46% male, 54% female	Developed their own questionnaire based on scales previously used in the literature. *Questions from the green values scale*—*6 statements about environmental beliefs.*	Factor analysis	2	S1: Positive perceivers—62.3% S2: Negative perceivers—37.7%
Keller et al.; 2019; Azerbaijan	N/R	419; 26.33 (3.18); 100% female ^c^	Three Factor Eating Questionnaire (modified). *Questions under 3 factors (emotional eating, cognitive control, uncontrolled eating).*	Cluster analysis; Ward’s method	3	S1: Functional eaters—36.6% S2: Conscious eaters—27.2% S3: Emotional and hedonic eaters—36.5%
Kitunen et al.; 2019; Australia	The Motivation, Opportunity, and Ability (MOA) theoretical framework	327; 20–35 (range) ^b^; 22.9% male ^c^, 77.1% female ^c^	Developed their own questionnaire based on scales previously used in the literature. *Questions about education level, motivation, ability, opportunity, and Turconi eating behaviour score.*	Cluster analysis; 2 step	2	S1: Breakfast skippers—48.6% S2: Weight conscious—51.4%
Koksal; 2019; Lebanon	N/R	411; <30–45+ (range); 49.6% male, 50.4% female	Developed their own questionnaire including the food choice motive scale which was previously used in the literature. *Questions under 8 food choice motives (ecological, sensory, convenience and availability, health. weight, mood, price, religion).*	Cluster analysis; 2 step	4	S1: Careless—14.3% S2: Conscious—35.0% S3: Hedonic—20.6% S4: Health and weight—29.9%
Lara et al.; 2014; United Kingdom	N/R	206; 61 (7); 40% male ^a^, 60% female ^a^	Perceived barriers to healthy eating.	Cluster analysis; 2 step	3	S1 ^d^—21.0% S2 ^d^—46.5% S3 ^d^—32.5%
Liu et al.; 2020; China	Means-end chain theory	438; 69.5 (6.9); 47.7% male, 52.3% female	Developed their own questionnaire. *Questions under 5 factors (food safety beliefs, light product interest, food taste beliefs, food freshness beliefs, general health interest).*	Cluster analysis; 2 step	3	S1: Health and safety concerned—38.6% S2: Hedonic and less health concerned—29.4% S3: Less safety and somewhat health concerned—32.0%
Milosevic et al.; 2012; Western Balkans	N/R	2813; 45.9; 48.2% male, 51.8% female	Food Choice Questionnaire. *Questions under 8 factors (health and natural content, mood, preparation convenience, purchase convenience, sensory appeal, price, weight control, familiarity and ethical concern).*	Cluster analysis; 2 step	5	S1: Food enthusiasts—23.1% S2: Unconcerned food consumers—22.2% S3: Price oriented and distressed—21.0% S4: Purchase convenience—17.2% S5: Health oriented—16.5%
Montero-Vicente et al.; 2019; Spain	N/R	500; 25–74 (range); N/R	Food-Related Lifestyle instrument (modified). *Questions under 5 factors (interest in cooking, interest in natural products, quality/price ratio, extra-domestic and social consumption, interested in nutrition and innovation).*	Cluster analysis; Ward’s method	4	S1: Total indifference—4.0% S2: Little time to cook, concerned about nutrition and extra-domestic consumption—26.4% S3: Cooks and preference for natural products—40.2% S4: Unconcerned—29.4%
Naughton et al.; 2017; Ireland	Transtheoretical Model of Behaviour Change; Theory of Planned Behaviour	477; 18–65+ (range) ^b^; 50.0% male; 50.0% female	Developed their own questionnaire based on scales previously used in the literature. *Questions under 6 social cognitive factors (confectionery consumption, hedonic hunger, perceived behavioural control, dietary planning, perceived need, lifestyle goal).*	Latent class analysis	4	S1: Triers—20.0% S2: Successful actors—17.0% S3: Thrivers—28.0% S4: Unmotivated—35.0%
Pentikainen et al.; 2018; Finland and Germany	Self-Determination Theory	Finland: 1060; 18–74 (range); 52.3% male, 47.7% female Germany: 1070; 18–74 (range); 49.7% male, 50.3% female	Three-Factor Eating Questionnaire (modified). *Questions under 3 factors (emotional eating, cognitive restraint, uncontrolled eating).*	Cluster analysis; 2 step	4	S1: Susceptible—Finland (20.4%), Germany (19.8%) S2: Easy going—Finland (30.8%), Germany (29.8%) S3: Rational—Finland (29.5%), Germany (31.3%) S4: Struggling—Finland (19.3%), Germany (19.1%)
Rejman et al.; 2019; Poland	N/R	600; 18–65+ (range); 38.3% male, 61.7% female	Developed their own questionnaire. *Questions from 14 food choice determinants (e.g., price, taste, nutritional value).*	Cluster analysis; k-means	3	S1: Non-Adopters—17.0% S2: Emergents—32.0% S3: Adopters—51.0%
Saba et al.; 2019; Italy	N/R	1224; 36.9 (12.8); 39.0% male, 61.0% female	Health and Taste Attitudes Scale (HTAS). *Questions under 3 sub-dimensions of the HTAS (general health interest, light product interest, natural product interest).*	Latent class cluster analysis	3	S1: Low health interest—28.2% S2: Medium health interest—53.4% S3: High health interest—18.4%
Sarmugam et al.; 2015; Australia	N/R	530; 49.2 (16.6); 41.7% male, 58.3% female	Developed their own questionnaire based on scales previously used in the literature. *Questions from two scales (impulse buying and food involvement).*	Cluster analysis; 2 step	3	S1: The impulsive, involved consumers—33.4% S2: The rational, health conscious consumers—39.2% S3: The uninvolved consumers—27.4%
Schnettler et al.; 2017; Chile	N/R	372; 20.4 (2.4); 43.5% male, 56.5% female	Developed their own questionnaire based on scales previously used in the literature. *Questions from 3 scales (satisfaction with food-related life scale, food technology neophobia scale, and food neophobia scale).*	Cluster analysis; 2 step	3	S1: Eating is of little relevance to their families—24.2% S2: Pressured to eat—25.0% S3: Enjoy the cohesiveness of family eating—23.9% S4: Eating is very important to their family—26.9%
Schnettler et al.; 2017; Chile	N/R	372; 20.4 (2.4); 43.5% male, 56.5% female	Developed their own questionnaire based on scales previously used in the literature. *Questions from 3 scales (satisfaction with food-related life scale, satisfaction with life scale, and food neophobia scale).*	Cluster analysis; 2 step	3	S1: Neophobic, satisfied with their food-related life—57.8% S2: Non-neophobic, satisfied with their food-related life—28.5% S3: Food neophobic, unsatisfied with their food-related life—13.7%
Schnettler Morales et al.; 2016; Chile	N/R	372; 20.4 (2.4); 43.5% male, 56.5% female	The Family Eating Habits Questionnaire (FEHQ). *Questions from 3 components (importance of eating to family members, cohesiveness of family eating, pressure to eat).*	Cluster analysis; 2 step	3	S1: Neophobics satisfied with their life and their food-related life—26.9% S2: Neophobics moderately satisfied with their life and their food-related life—40.8% S3: Non-neophobics satisfied with their life and their food-related life—32.3%
Simunaniemi et al.; 2013; Sweden	N/R	1191; 18–64 (range)^b^; 56.0% male^a^, 44.0% female ^a^	Developed their own questionnaire based on scales previously used in the literature. *Questions from 5 factors (determinants of fruit and vegetable consumption, habit, perceived barriers, perceived physical environment, knowledge).*	Cluster analysis; 2 step	2	S1: Positive cluster—40.0% S2: Indifferent cluster—60.0%
Voinea et al.; 2019; Romania	N/R	1185; 18–65+ (range); 35.7% male, 64.3% female	Developed their own questionnaire. *Questions under 5 factors (interest in a healthy diet, importance of taste and other sensory characteristics in choosing consumed foods, importance of the nutritional value of the diet, the degree of blog use that contain topics about healthy eating, and BMI).*	Cluster analysis; 2 step	2	S1: Interested—57.5% S2: Eclectics—42.5%
Wetherill et al.; 2018; USA	Hierarchy of Food Needs Model	73; 41.5 (14.9) ^b^; 27.2% male ^a^, 72.8% female ^a^	Food Choice Values Questionnaire (modified). *Questions under 8 factors (convenience, accessibility, tradition, comfort, organic, safety, sensory appeal, weight control/health).*	Cluster analysis; 2 step	4	S1: Limited endorsement of food choice values—23.0% S2: Safety and sensory—33.0% S3: Health and weight control—18.0% S4: Broad endorsement of many food choice values—26.0%

BMI: Body Mass Index, SES: socio economic status, FYRoM: Former Yugoslav Republic of Macedonia. N/R: not reported S1: Segment 1 and so on. ^a^ studies that reported sex—not gender. ^b^ demographics (i.e., age & gender/sex) reported for the total sample rather than the number of people involved in segmentation. ^c^ gender/sex is not differentiated so classified as N/R. ^d^ segments did not have names. ^e^ percentages reported as in the paper and do not add up to 100%.

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
