# Peer review of "Psycho-Behavioural Segmentation in Food and Nutrition: A Systematic Scoping Review of the Literature"

_nutrients, 2021, doi:10.3390/nu13061795_

Round 1

Reviewer 1 Report

I have mixed reviews about this paper. The title indicates a very significant area of study but as you read through the second half of the introduction, the contents become hazy and challenging to follow through. Furthermore, the aim of the scoping review gets lost and as a reviewer, I begin seeing elements of a systematic review rather than a scoping review.

Author Response

  • I have mixed reviews about this paper. The title indicates a very significant area of study but as you read through the second half of the introduction, the contents become hazy and challenging to follow through. Furthermore, the aim of the scoping review gets lost and as a reviewer, I begin seeing elements of a systematic review rather than a scoping review.

RESPONSE: We thank you for your time and comments to help us improve our paper. We have addressed your feedback about the introduction (see below) and believe it will now be easier to follow and clearer to understand. All line numbers relate to the tracked changes document. 

Our scoping review methodology was conducted in line with the Equator Network guidelines (https://www.equator-network.org/reporting-guidelines/prisma-scr/) and therefore we would call it a ‘systematic scoping review’ and have amended the title of the paper. As outlined in the guidelines and Munn et al., the difference between a scoping and systematic review lies in the aims and outcomes of the paper (See https://bmcmedresmethodol.biomedcentral.com/articles/10.1186/s12874-018-0611-x). Our review is a scoping review because we were not evaluating the clinical effectiveness or assessing feasibility of an intervention, nor did we have a clear outcome required for inclusion in the review. Instead we had a broad, exploratory focus - and aimed to give an overview of the research that has conducted psycho-behavioural segmentation in food and nutrition, as well as explore the included papers characteristics such as the methods used (i.e. post-hoc or a-priori) and the differences in study type. Our results are reported under key subheadings (e.g. method of segmentation) to explore the concepts further and meet our aim of providing an overview of the research area.. We have added Munn et al., to the text to clearly state that a scoping review was chosen because of the aim of the paper - “The overarching aim of this paper is to provide formative research to determine the breadth and scope of segmentation within the contemporary food and nutrition environment. A systematic scoping review was deemed most appropriate for this purpose (Munn et al)” Line 131.

  • Line 35-change lead to led.

RESPONSE: Thank you, we have corrected this.

  • Line 36-39- fragment the sentence to read better.

RESPONSE: Thanks for the suggestion. This sentence has been revised into two sentences: “Contemporary living environments contribute to poor dietary choices by increasing the accessibility and exposure to inexpensive, energy dense, nutrient poor but highly palatable food and beverages. These environments, paired with technology advances, transport, and community structures that decrease physical activity have further contributed to poor health outcomes.” Line 36-40.

  • I find the introduction section long and monotonous, lacking a clear rational since most of the text in the 2nd half of the introduction (segmentation base) is duplicated in the Table 1- Remove the repetitive phrases that are duplicated in Table 1 and adopt a more complementary approach. 

RESPONSE: Thank you for your feedback. We agree that the introduction was too long. We have amended the structure to include Table 1 as a supplementary file rather than have it in-text. Furthermore, we have edited the last paragraph in the introduction (Lines 95- 119) to be more concise, moving the examples in practice to be in Supplementary File 1.

  • Line 117 and 118 is unnecessary since the approach is from a social marketing lens-consider deleting it. 

RESPONSE: Thank you for this suggestion. This sentence has now been deleted.

  • I find a section on the study quality missing, and I recommend that it is revised and inserted between data extraction and summary of findings 

RESPONSE: A quality appraisal, whilst insightful, is not a requirement for a scoping review and that is why it is not included. Our review does not ask a question about the effectiveness of an intervention/trial, or to be precise the effectiveness of psycho-behavioural segmentation, but rather aims to scope the topic area and find out what has been done so far. In the reference you have provided below (Arksey and O’Malley, 2005) they mention  -  “the scoping study does not seek to assess quality of evidence and consequently cannot determine whether particular studies provide robust or generalizable findings”. Therefore, as we have not tried to generalise the included paper’s findings, we do not think a quality appraisal would change our results or reporting. Therefore we have chosen not to add a quality assessment to our scoping review.  In the limitations section we had highlighted that this is a methodological limitation of undertaking a scoping review (Line 437-440).

  • Replace summary of findings with just results 

RESPONSE: This subheading has been amended to ‘Results’ Line 174.

  • I feel that the theory section should come before the findings since they contextualize the results 

RESPONSE: Traditionally the format has been to summarise the parameters that are consistent across studies prior to reporting nuances in study protocols and findings. Thus, we feel it is important to state the number of inclusions and their key characteristics prior to reporting on the theory section, particularly as only nine included studies used a theory and therefore the majority of the papers did not have one. 

  • I find the reporting of the results not clear and satisfactory in the sense that the aim of the study gets lost in between. Furthermore, the objective of a scoping review is to map the existing literature and identify gaps to inform practice but, I find the paper falls short of this. Even though in the discussion section the authors highlight the gaps in research (See Arksey and O’Malley, 2005).

RESPONSEWe are unsure how to action your comments about the results not being satisfactory. The aim of the scoping review was to explore and summarise the existing literature related to psycho-behavioural segmentation in food and nutrition. To meet this aim, we have reported key characteristics of each study, theories incorporated, the method of segmentation they used, the tool used for segmentation, and the outcomes of segmentation (i.e. the segments generated) all under different subheadings to ensure clarity. We feel that this provides a broad overview of the research (while focusing on some key aspects) and therefore meets our aim. We have also edited the discussion to initially restate the aim of the study and repeat our key findings so that the aim does not get lost (Line 282-289). The majority of the discussion relates to practical actions i.e. considering different approaches to segmentation other than what has commonly been done so far and therefore we are also unsure how to action your comment of our review falling short.

Reviewer 2 Report

For an article that is based on a literature review, that is, on research by other scientists, it is quite neat.

The article requires a slight linguistic correction.

For an article that is based on a literature review, there are too few items from the last two years. Generally, there is little literature that the authors have researched / pointed out. Usually, articles based on a literature review have more than 100 items.
The introduction is too long, it should not exceed 2 pages - the table should be moved to another section.
The source should be placed under the drawing: is it your own or a borrowed study.
Table 2 would be clearer to group articles from different countries by continents (Europe, Asia, North and South America etc.) - the layout should be proportional, not 5 articles from Chile and none from Brazil.
Make the Limitation and Future Research sections as a separate point or add to the conclusion.

Author Response

  • For an article that is based on a literature review, that is, on research by other scientists, it is quite neat. The article requires a slight linguistic correction.

RESPONSE: We would like to thank you for your time and feedback on our manuscript. Unfortunately, we are unsure what you mean by the linguistic correction hence are unable to action it. If there are linguistic errors, they will be addressed in the copy-editing phase.

  • For an article that is based on a literature review, there are too few items from the last two years. Generally, there is little literature that the authors have researched / pointed out. Usually, articles based on a literature review have more than 100 items.

RESPONSEThanks for your interesting perspective. We have reviewed our references to determine how current they are and have counted that 40 out of our 68 references (59%) were published in the last 5 years. Furthermore, 9 of 68 (13%) of our references are from 2020 and 2021, despite the extensive impacts on the publishing industry from COVID-19 which has significantly delayed such research. 

Our scoping review methodology was conducted in line with the Equator Network guidelines (https://www.equator-network.org/reporting-guidelines/prisma-scr/) whereby we defined the population, concept and context to ensure rigor and transparency of the methods and ensure the results are trustworthy (See Munn et al., https://bmcmedresmethodol.biomedcentral.com/articles/10.1186/s12874-018-0611-x). Therefore, the discrepancy you mention of having too few items may be because psycho-behavioural segmentation research is not common (hence our review idea) and therefore there are less papers about this topic than you may see in other reviews (for example about cardiovascular disease). 

  • The introduction is too long, it should not exceed 2 pages - the table should be moved to another section.

RESPONSE: Thank you for your helpful suggestions. We agree that the introduction was too long. We have moved Table 1 to be a Supplementary file based on both your comments and comments from Reviewer 1. Furthermore, to make the introduction more concise, we have moved all examples of segmentation in practice from the last paragraph into Supplementary File 1 (See tracked changes document for edits lines 97-122). The length is now within 2 pages. 

  • The source should be placed under the drawing: is it your own or a borrowed study.

RESPONSE: This is a PRISMA flow diagram, included in the equator network guidelines (https://www.equator-network.org/reporting-guidelines/prisma-scr/). We have amended the figure legend to reference the PRISMA statement to make this clearer: “Figure 1. Preferred Reporting Items for Systematic Review (PRISMA) Scoping Review flow diagram, adapted from the 2020 PRISMA statement [24]” Line 163-164.

  • Table 2 would be clearer to group articles from different countries by continents (Europe, Asia, North and South America etc.) - the layout should be proportional, not 5 articles from Chile and none from Brazil.

RESPONSEAs we defined the population, concept and context to ensure rigor and transparency of the methods and ensure the results are trustworthy (See link to Munn et al., above), our results are not proportional. Therefore we cannot change the layout of our table. The inclusions were not hand-picked and we did not use stratified sampling to ensure research was included from certain countries. It just happens that this type of research is more common in certain regions.  We have included the country in Column 1 of the results table (now Table 1) to demonstrate this discrepancy.

  • Make the Limitation and Future Research sections as a separate point or add to the conclusion.

RESPONSE: The limitations and future directions are placed in the discussion section as per the journal formatting guidelines. We have followed the instructions to authors and therefore cannot change this section to be in the conclusion. As they currently stand, they are separate points due to the use of subheadings (i.e. limitations is section 4.3 and future directions is 4.4) and so we are unsure how to separate this more as you have suggested, whilst still following the journal instructions.

Round 2

Reviewer 1 Report

The authors have adequately responded to my previous concerns. I find that by them revising the title to "systematic scoping review", It has made the methodology and the result sections clearer. 

Minor corrections: please correct the word spacing in the entire document and format the tables i.e. Table 1

Author Response

Thank you again for your time and comments. 

We have re-formatted the table to be in portrait mode and have removed some double-spacing that was throughout the document i.e. lines 209, 214, 345, 432 tracked changes document. Please let us know if we have misinterpreted your comment.  As far as we can tell the other formatting such as line spacing and paragraph spacing is consistent with the instructions to authors.